# Biosynthesis of Sulfur-Containing Small Biomolecules in Plants

**DOI:** 10.3390/ijms21103470

**Published:** 2020-05-14

**Authors:** Yumi Nakai, Akiko Maruyama-Nakashita

**Affiliations:** 1Department of Biochemistry, Osaka Medical College, 2-7 Daigakumachi, Takatsuki 569-8686, Japan; 2Department of Bioscience and Biotechnology, Faculty of Agriculture, Kyushu University, 744, Motooka, Nishi-ku, Fukuoka 819-0395, Japan; amaru@agr.kyushu-u.ac.jp

**Keywords:** sulfur-containing small biomolecules, cysteine desulfurase: sulfane sulfur, iron-sulfur (Fe/S) cluster, molybdenum cofactor (Moco), rhodanese (RHD), sulfur modification of tRNA, persulfide

## Abstract

Sulfur is an essential element required for plant growth. It can be found as a thiol group of proteins or non-protein molecules, and as various sulfur-containing small biomolecules, including iron-sulfur (Fe/S) clusters, molybdenum cofactor (Moco), and sulfur-modified nucleotides. Thiol-mediated redox regulation has been well investigated, whereas biosynthesis pathways of the sulfur-containing small biomolecules have not yet been clearly described. In order to understand overall sulfur transfer processes in plant cells, it is important to elucidate the relationships among various sulfur delivery pathways as well as to investigate their interactions. In this review, we summarize the information from recent studies on the biosynthesis pathways of several sulfur-containing small biomolecules and the proteins participating in these processes. In addition, we show characteristic features of gene expression in *Arabidopsis* at the early stage of sulfate depletion from the medium, and we provide insights into sulfur transfer processes in plant cells.

## 1. Introduction

Versatile chemical reactivity of sulfur in nature is widely recognized to be biologically significant for living organisms [1]. In plants, sulfur is an essential macronutrient required for growth, a significant fraction of which is incorporated into proteins [2]. Sulfur contents in plants depend on the availability of sulfur in the environments but, in some cases, reach 0.3% of their dry weight [3]. Plants take up sulfate from roots through the function of several sulfate transporters (*SULTR*s) and use it as a starting material for sulfur assimilation [2]. Sulfate is converted to 5’-adenylyl sulfate (APS) and reduced to sulfite by APS reductase (APR). Sulfite is then reduced to a sulfide ion by sulfite reductase (SiR), and the resultant sulfide reacts with *O*-acetyl-L-serine to form cysteine [2]. While the conversion from sulfate to sulfide occurs in plastids, the subsequent cysteine biosynthesis requires coordinated functions of plastids and mitochondria as well as the cytosol [3]; *O*-acetyl-L-serine which is predominantly provided from mitochondria together with sulfide supplied from chloroplasts serve as substrates for bulk cysteine biosynthesis in the cytosol [3]. The expression of several *SULTR*s genes in roots and the *APR1*, *APR2,* and *APR3* genes is enhanced in sulfur-limited growth conditions [4,5,6,7,8,9].

Cysteine is the final product of sulfur assimilation and a key molecule in intracellular sulfur transfer processes (Figure 1) [10]. The cysteinyl thiol groups in proteins are important for the maintenance of their tertiary structures through disulfide bond formation [11]. Cysteinyl thiols also play essential roles in controlling the redox states of redox proteins, such as glutaredoxins (Grxs) and thioredoxin (Trxs) [11]. Furthermore, the thiol group in cysteine can be a ligand for metal ions or metal-containing cofactors. On the other hand, free cysteine does not act as a redox factor in cells because the thiol group of free cysteine is highly reactive and can form undesirable substances or by-products, and can even be cytotoxic under physiological conditions in which it is present in high concentrations [11,12]. On the contrary, the thiol group of glutathione (gamma-glutamyl-cysteinyl-glycine, GSH), which is an abundant and essential intracellular tripeptide thiol, has low cytotoxicity [12]. GSH plays versatile roles to maintain the intracellular redox states and is synthesized from cysteine by two consecutive steps [3,13].

Cysteine is also a major sulfur donor in biosynthetic processes of a variety of small sulfur-containing small biomolecules, such as Fe/S clusters, molybdenum cofactor (Moco), and sulfur-containing nucleosides of tRNAs (Figure 1) [14]. In plants, these sulfur-containing small biomolecules are necessary for maintaining cellular homeostasis, including respiration, photosynthesis, and various primary and secondary metabolisms. In the biosynthetic processes of sulfur-containing small biomolecules, sulfur from cysteine is sequentially transferred among various sulfur-carrying proteins as sulfane sulfur in the form of persulfide groups of proteins [15,16]. However, many details underlying the control and regulation of the sulfur transfer processes of sulfur-containing small biomolecules in plant cells are not fully understood. Elucidation of the biosynthetic pathways of sulfur-containing small biomolecules and their interactions is a prerequisite for the comprehensive understanding of intracellular sulfur transfer processes. In this review, we focused on recent studies concerning the biosynthetic pathways of sulfur-containing small biomolecules in plant cells, exemplified by Fe/S clusters, Moco, and sulfur-modified tRNAs, and we analyzed their interconnections and associated intracellular sulfur transfer processes. By analyzing the early responses of gene expression in *Arabidopsis* plants exposed to sulfate-depleted conditions, we report new insights into sulfur delivery to these biomolecules in plant cells.

## 2. Fe/S Cluster Biosynthesis in Plant Organelles

Fe/S clusters are prosthetic groups consisting of acid-labile sulfur and non-heme iron that are incorporated into various apoproteins to form so-called “Fe/S proteins”. Mitochondria and plastids have their own pathways for Fe/S cluster biosynthesis, both of which involve the organelle-specific L-cysteine desulfurases NFS1 and SUFS, respectively (Figure 2). L-cysteine desulfurase (EC 2.8.1.7) is a pyridoxal-5’-phosphate-containing protein that is ubiquitously found in all phyla. L-cysteine desulfurase catalyzes the reaction that removes a sulfur atom from the L-cysteine substrate to produce L-alanine. During this catalytic reaction, L-cysteine desulfurase transiently binds the sulfur derived from the L-cysteine as a sulfane sulfur in the form of persulfide, after which the sulfur bound to the enzyme is transferred to an Fe/S biosynthetic “scaffold protein” or its complex. Mitochondria and plastids have their own scaffold proteins in which an unstable nascent Fe/S cluster is formed. The nascent unstable Fe/S clusters are then assembled and maturated with the assistance of additional Fe/S carrier proteins and chaperones, and finally incorporated into the apo-form of the target proteins (Figure 2). Both the mitochondrial nitrogen fixation protein for sulfur transfer (NFS) pathway and the plastidial sulfur utilization factor (SUF) pathway are indispensable for these energy-producing organelles because many Fe/S proteins play essential roles in the electron transfer systems of respiration and photosynthesis [17,18,19,20,21].

In mitochondria, NFS1 functions as a sulfur donor, and ISD11 is thought to stabilize NFS1 [21]. Sulfur that transiently binds to NFS1 is incorporated into the mitochondrial scaffold ISU proteins to form an unstable Fe/S cluster [22,23]. Three ISU proteins have been identified in *Arabidopsis* mitochondria, among which *ISU1* seems to be a major scaffold protein because the expression levels of other two ISU proteins (*ISU2* and *ISU3*) are very low [21,22,24]. It has also been reported that ISA1 and ISCA4 contribute to the formation of mitochondrial Fe/S clusters, possibly as carrier proteins [25]. Molecular chaperones HSCA1, HSCA2, and HSCB are also thought to assist mitochondrial Fe/S cluster biosynthesis [26]. In addition, several other proteins are needed to form distinct types of Fe/S clusters or Fe/S proteins; NFU5/NFU-I and NFU4/NFU-III are required to form [4Fe-4S]-type Fe/S clusters [18,27]. A recent report showed that these NFU proteins are reduced by the Trx-mediated redox system, rather than the Grx-mediated system [28]. IBA57 and INDH are proteins involved in the maturation of the Fe/S cluster in mitochondrial respiratory chain complex I [25,29]. It has also been reported that mitochondrial monothiol glutathione oxidoreductase GRXS15, is required for Fe/S cluster biosynthesis [30]. While many protein factors required for mitochondrial Fe/S biosynthesis have already been identified and characterized, their precise biochemical functions and functional cooperation are not yet fully characterized. Moreover, the molecular machinery required for the formation and incorporation of distinct types of Fe/S clusters into various apoproteins still remains unclear.

As for *Arabidopsis* plastidial Fe/S cluster biosynthesis pathway, based on the analogy to the known bacterial SUF system [31], it is hypothesized that SUFS provide sulfane sulfur to a scaffold complex that consists of SUFB, SUFC, and SUFD, and contributes to the initial biosynthetic reaction [32,33]. SUFE1 cooperates with SUFS and is thought to be involved in sulfur mobilization from cysteine [34,35]. Two other SUFE-like proteins, *SUFE2* and SUFE3, can also activate SUFS in vitro [36], but their in vivo functions remain to be determined. In addition, many other proteins are also known to be involved in the later steps of plastidial Fe/S cluster formation and/or delivery, including the ISCA-type protein SUFA [37,38]; three nitrogen-fixation-subunit-U (NFU)-type proteins CNFU1/NFU1, CNFU2/NFU2, and CNFU3/NFU3 [18,27,39,40,41,42]; P-loop nucleotide phosphatase HCF101 [43]; IBA57.2 [20,44]; and monothiol Grx-like proteins such as GRXS14 [45].

## 3. Maturation of the Fe/S Cluster in the Cytosol and Related Proteins in Both Mitochondria and the Cytosol

In both mitochondria and plastids, biosynthesized Fe/S clusters can be incorporated into various Fe/S proteins that function inside respective organelles. However, there are also many Fe/S proteins that exist outside of these organelles that play important roles in various redox and metabolic processes. As is found in the yeast case [46,47], mitochondrial NFS1 is required for Fe/S protein synthesis outside of mitochondria also in *Arabidopsis* [21]; however, the exact sulfur donor for the cytosolic Fe/S cluster has not yet been elucidated. ATM1 is a highly conserved mitochondrial ATP-binding cassette transporter that is also involved in the cytosolic Fe/S cluster biosynthesis [25,48,49]. *Arabidopsis* possess three ATM1-like proteins [50,51] and ATM3 is involved in the Fe/S cluster maturation in cytosol [52], however, the exact mechanism that the Fe/S cluster or its constituents are transported from mitochondria to the cytosol via ATM3 still remain to be elucidated yet.

In addition to the organelle-located proteins we have discussed, the cytosolic Fe/S cluster assembly pathway (CIA pathway) in plants contains another set of cytosolic proteins, similar to what has been reported in yeast [47]. In *Arabidopsis*, the CIA pathway includes NBP35, but does not include a homolog for the yeast Cfd1-like protein [53,54]. DRE2, which can interact with TAH18/CIAPIN, is suggested to function as an electron donor for the assembly of the Fe/S cluster in the cytosol of plant cells [55,56], and a recent report suggested that AtNEET, a NEET-like protein involved in iron metabolism in plastids, can interact to the cytosolic DRE2 [57,58,59]. Cytosolic proteins such as NAR1, CIA1, CIA2/AE7, and MET18 are also involved in the maturation of the Fe/S cluster in the cytosol [60]. Moreover, GRXS17, a unique Grx family protein characterized by its N-terminal Trx-like domain, interacts with CIA components [61]. Interestingly, GRXS17 is involved in the auxin response during plant development [62,63], as well as in redox regulation in response to iron-deficient conditions [64]. However, a direct interaction between GRXS17 and proteins related to the auxin or iron deficiency response remains unknown [65].

## 4. Biosynthesis of Cytosolic Sulfur-Containing Small Biomolecules Other Than Fe/S Clusters

In addition to the Fe/S protein, the cytosol contains other sulfur-containing small biomolecules, such as Moco and sulfur-modified tRNAs (Figure 1) [14]. In the biosynthesis of sulfur-containing small biomolecules in the cytosol, it is necessary to ensure that sulfur is not damaged by cytosolic oxidizing factors and is transported without being used in the cytosolic redox regulation system. As described below, a protein-bound persulfide group in rhodanese (RHD) domain-containing proteins might be the most suitable system for carrying and transferring labile sulfur in the cytosol. Such sulfur transfer systems are found in both the Moco biosynthesis pathway and sulfur-modification processes of certain tRNAs.

Moco is a prosthetic group in which molybdenum is chelated to sulfur atoms bound to the pterin-based frame, molybdopterin (MPT) (Figure 3). In plants, nitrate reductase, xanthine dehydrogenase, and aldehyde oxidase belong to the Moco enzyme family [66]. The Moco biosynthetic pathway is widely found in bacteria such as *Escherichia coli* as well as many eukaryotes like plants and humans; however, it is not found in some fungi, including *Saccharomyces cerevisiae*, and protozoa [67,68]. Early steps of eukaryotic Moco biosynthesis occur in mitochondria, where GTP is cyclized by catalyzing the reaction between the mitochondrial Fe/S protein CNX2 and CNX3 in order to form cyclic pyranopterin monophosphate (cPMP) [66,69]. Subsequently, cPMP is exported to the cytosol from mitochondria, at which point two sulfur atoms are incorporated into cytosolic cPMP to form MPT. In this step, sulfurtransferase CNX5/STR13 interacts with CNX7 with the aid of CNX6 (Figure 3).

*Arabidopsis* CNX5/STR13 is a unique sulfurtransferase consisting of a MoeB-like domain at the amino terminal region and a RHD domain at its carboxyl terminal region [67,70,71]. From the structural similarity and substrate specificity among eukaryotic and bacterial Moco biosynthesis proteins [72,73,74], *Arabidopsis* CNX7 is thought to initially activated by binding to the MoeB-like domain of CNX5 [67], while the sulfur binds to the RHD domain as a persulfide of CNX5. After that, an intermolecular sulfur transfer between the persulfide-derived sulfur and CNX7 seems to occur on CNX5.

CNX5 plays a pivotal role also in sulfur modification of cytoplasmic tRNAs [67]. Sulfur modification of tRNA is ubiquitously found at the first anticodon uridine (i.e., the wobbleU) of tRNA^Lys^_UUU_, tRNA^Glu^_UUC_, and tRNA^Gln^_UUG_ [75], and many proteins participate in this modification process [14,67]. Human MOCS3 and yeast Uba4, both of which are orthologous to the plant CNX5, add sulfur to the C-terminal glycine residue of the ubiquitin-related modifier (URM) protein Urm1 to form a thiocarboxylate [76,77] in the process of the sulfur modification of tRNAs. Thus, also in *Arabidopsis*, CNX5 can add sulfur to the plant URM proteins (URM11 and URM12) in a similar manner to the human and yeast cases [67]. That is, CNX5 is a key protein for sulfur delivery in the cytosol because it interacts with URM11 and URM12 in tRNA sulfur modification of the wobbleU, as well as interacting with CNX7 in MPT formation during Moco biosynthesis. However, the exact sulfur donor molecule or protein(s) to the cytosol CNX5 has not been elucidated at present (shown as X in Figure 3).

ABA3 is another protein with a two-domain structure, one of which binds to a substrate protein and the other RHD domain transiently binds sulfur. ABA3 is an enzyme required for the formation of sulfur-modified Moco (S-Moco), a cofactor of aldehyde oxidase and xanthine oxidase. ABA3 binds to Moco at the C-terminal domain and moves sulfur atoms from a persulfide temporarily formed on the RHD-like domain to the bound Moco in order to form a S-Moco molecule [78,79]. Proteins, which possess a substrate activation domain together with a RHD domain that binds sulfane sulfur in the form of persulfide, may be advantageous for the delivery of sulfane sulfur in the construction of cytosolic sulfur-containing small biomolecules.

## 5. Gene Expression Dynamics of the Biosynthesis of Sulfur-Containing Biomolecules in Sulfate-Deprived Roots

Investigation of the initial response of plant root cells to sulfate depletion may help to identify proteins that alter the transcript levels in response to the changes in intracellular sulfur flux. Therefore, using previously reported microarray data of gene expressions in *Arabidopsis* roots transferred to the sulfate-depleted medium [80], we re-analyzed the data and searched for the sulfur-depletion responsive genes involved in the above-mentioned sulfur transfer pathways.

Interestingly, we found that the both the gene expressions of the scaffold proteins *ISU2* and *ISU3* slowly increase 12 h after sulfate depletion (Figure 4).

It is conceivable that different scaffold proteins with different affinities for various substrate proteins would be needed to maintain Fe/S cluster biosynthesis in accordance with various sulfate concentrations in the environment and thus, the genes encoding scaffold proteins may show distinct expression responses. Considering that the expression of *ISU2* and *ISU3* is much lower than that of *ISU1* in normal growth conditions [22,24], it is possible that *ISU2* and *ISU3* have higher affinity for sulfur or specific client proteins than *ISU1*. If this is the case, both *ISU2* and *ISU3* might be more suitable scaffold proteins for binding a nascent Fe/S cluster than *ISU1* under the conditions where intracellular sulfur is limited, even though their expression is normally repressed in sulfur-replete conditions. A nascent Fe/S cluster preassembled on the scaffold might need to be immediately transferred to subsequent steps of biosynthesis in order to form Fe/S proteins. Otherwise, labile and exposed Fe/S clusters on scaffold proteins could easily react with oxygen to form reactive oxygen species, which can be cytotoxic. Under normal conditions where the sulfane sulfur is sufficiently supplied, the highly expressed isoprotein *ISU1* may have lower affinity for sulfur and would be enough to deliver such transient and unstable Fe/S clusters to the target apoproteins. In such a situation, scaffold proteins with high affinity to sulfur may not be suitable for binding a nascent cluster because they presumably tend to cause excess or prolonged retention of unstable Fe/S clusters on the scaffolds. It would, therefore, be interesting to investigate whether the biochemical properties of *Arabidopsis* ISU isoproteins are actually differentiated and if they are relevant for different physiological conditions in terms of sulfate availability.

The expression of the key enzymes for cysteine assimilation such as *SULTR*s (for example, *SULTR1;2* and *SULTR2;1*), and APR genes (*APR1*, *APR2*, and *APR3*), is significantly increased when sulfate is depleted from growth medium (Figure 4, [80]). Conversely, gene expression of cytosolic glutaredoxins (e.g., *GRXS6*) and other redox proteins such as mercaptopyruvate sulfurtransferase (e.g., *MST1* and *STR2/RDH2*) seems to be slightly suppressed in response to sulfate depletion.

In contrast, gene expression of other cytosolic proteins involved in the biosynthesis of various sulfur-containing small biomolecules does not seem to respond significantly in the early phase of sulfate depletion (Figure 4). In addition, for most of the Fe/S proteins, including ferredoxins (FD1-FD4, FDC1, and FDC2) and aconitases (ACO1, ACO2, and ACO3), or Moco enzymes (such as nitrate reductases, *NR1* and *NR2*, and aldehyde oxidases, *AAO1*, *AAO2*, and *AAO3*), their gene expression also does not seem to respond to sulfate depletion (Figure 4).

## 6. Perspectives

Cysteine, the final product of sulfur assimilation in plants, is an important amino acid with a thiol group, which is a unique constituent of proteins. In addition to other thiol-containing molecules like glutathione and sulfur-containing secondary metabolites like glucosinolates, cysteine is an important substrate for biosynthesis of sulfur-containing small biomolecules [9].

In plant cytosol, as described above, a unique sulfurtransferase CNX5 transfers sulfur to the partner protein of two distinct biosynthetic pathways (Figure 3). The MoeB-like activation domain of CNX5 binds to CNX7 or URM proteins (URM11 and URM12) at its active cysteine residue of CNX5, while sulfur binds to another cysteine residue in the RHD domain of CNX5 as persulfide sulfane sulfur. Binding and activation of the substrate requires an acyl-thio bond between the cysteine residue of the enzyme and substrate protein. This protein–protein interaction through acyl-thio bond formation is similar to that of the interaction between a ubiquitin-like protein and a ubiquitin activating enzyme-like protein pairs (i.e., UBL-UBA pairs), which are widely found in eukaryotes [67,81,82,83]. Thus, a series of intramolecular reactions of CNX5, involving the activation of a partner protein, the sulfane sulfur binding via RHD persulfide, and the subsequent transfer of sulfur to the partner protein, is an excellent system for the safe transport of sulfane sulfur in the cytosol. Many sulfurtransferases that contain one or two RHD domain(s) are present in *Arabidopsis* [84]. Similar to Cnx5, another unknown sulfurtransferase that can bind and activate the substrate and simultaneously form persulfide within the RHD domain might be able to participate in the formation of another sulfur-containing small molecules in the plant cytosol.

In plant cells, two independent pathways for Fe/S cluster biosynthesis occurred in mitochondria and plastids have similar ways of using and assembling sulfur into the Fe/S clusters. As nascent labile Fe/S clusters on the scaffold proteins are highly reactive and may even cause cytotoxicity when reactive oxygen species are formed, strictly regulated Fe/S clusters transfer to downstream target or carrier proteins is needed. Moreover, a regulatory mechanism for sulfur supply may be required to maintain restricted sulfur flow for the biosynthesis of nascent Fe/S clusters. Microarray data suggests that the expression level of mitochondrial *ISU2* and *ISU3* is likely regulated in response to environmental sulfate levels (Figure 4). For the plastidial Fe/S biosynthesis system, gene expression of *SUFE2*, which is a known partner protein for plastidial cysteine desulfurase, is slightly and transiently increased upon sulfate depletion, whereas those of other *SUFE* paralogs do not seem to be more significantly affected. However, it still remains unclear whether this is relevant for the regulatory response in plastidial Fe/S cluster biosynthetic pathway.

Therefore, further studies on the regulatory mechanisms of the formation of nascent Fe/S clusters in plant organelles, as well as on the function of cytosolic sulfurtransferase that contain a RHD-domain, are necessary to clarify the intracellular regulation of sulfur transfer processes, including small sulfur-containing biomolecules.

## Figures and Tables

**Figure 1 ijms-21-03470-f001:**
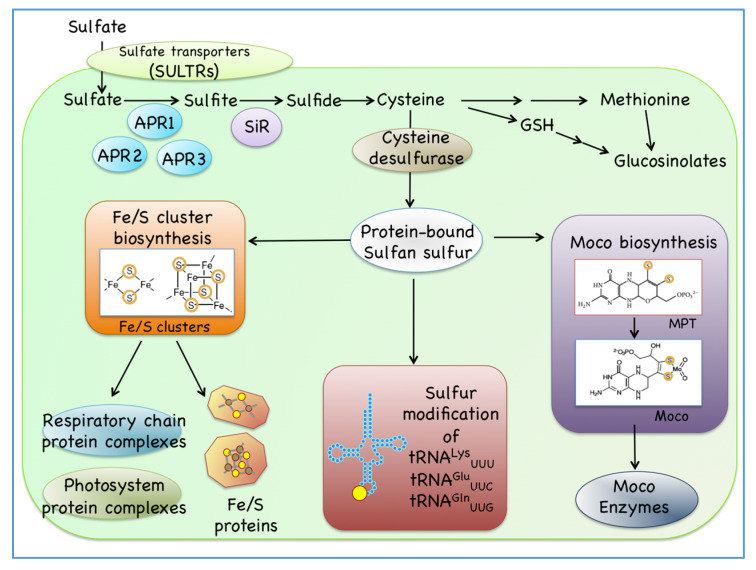
Cysteine is the end product of the sulfur assimilation and is required for the synthesis of various sulfur-containing small biomolecules such as iron-sulfur cluster (Fe/S), molybdenum cofactor (Moco), and sulfur-modified tRNAs in plant cells.

**Figure 2 ijms-21-03470-f002:**
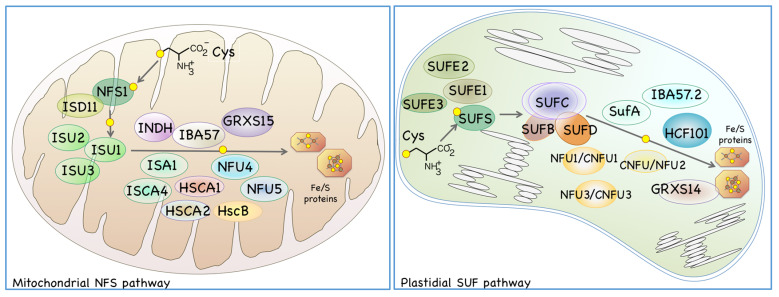
Two distinct Fe/S cluster biosynthetic pathways exist in mitochondria (mitochondrial NFS pathway, left) and plastid (plastidial SUF pathway, right) in *Arabidopsis*. Sulfur from the L-cysteine (Cys) substrate is assembled into a nascent Fe/S cluster of the scaffold proteins, and then the cluster is transferred through carrier proteins before finally being incorporated into various Fe/S apo-proteins. As examples, the Fe/S proteins of the [2Fe-2S] type and of the [4Fe-4S] type (iron ions and sulfur atoms are indicated as brown and yellow dots, respectively) are shown.

**Figure 3 ijms-21-03470-f003:**
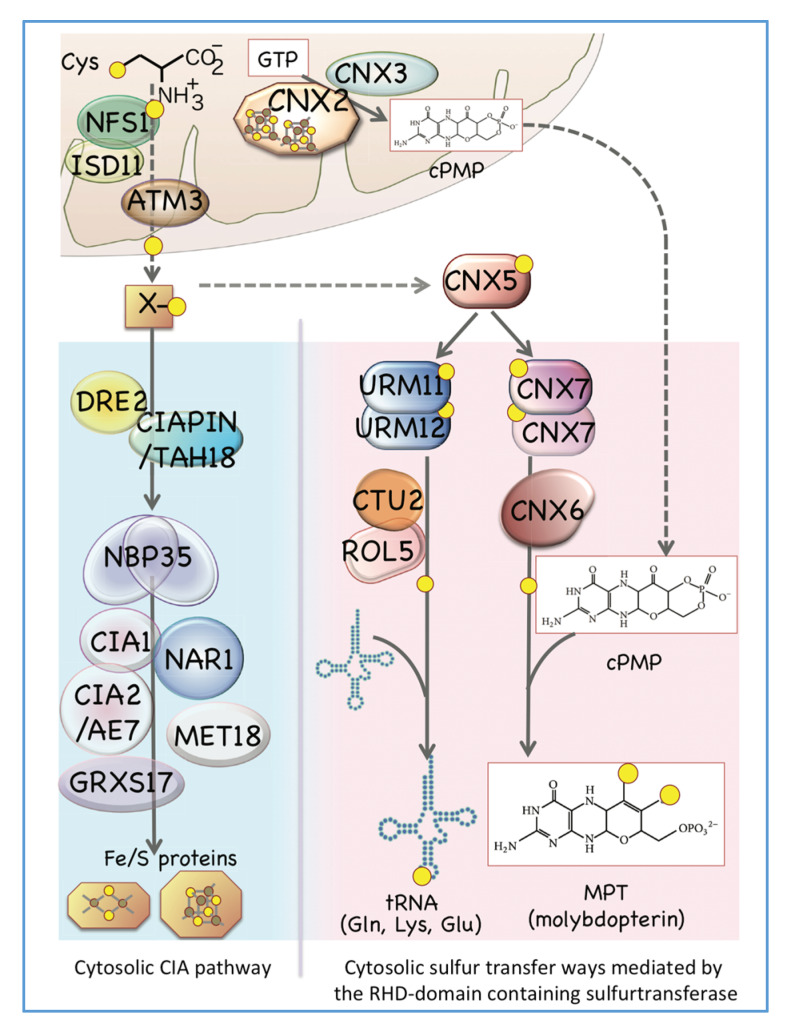
Two different sulfur transfer processes found in *Arabidopsis* cytosol. One is the cytosolic Fe/S cluster assembly pathway (CIA) pathway for the maturation of the Fe/S clusters (iron ions and sulfur atoms are indicated as brown and yellow dots, respectively) (left), and the other is the sulfur transfer pathway mediated by the rhodanese (RDH) domain-containing sulfurtransferase CNX5, which is required to the biosynthesis of molybdopterin (MPT) and sulfur-modified tRNAs (right).

**Figure 4 ijms-21-03470-f004:**
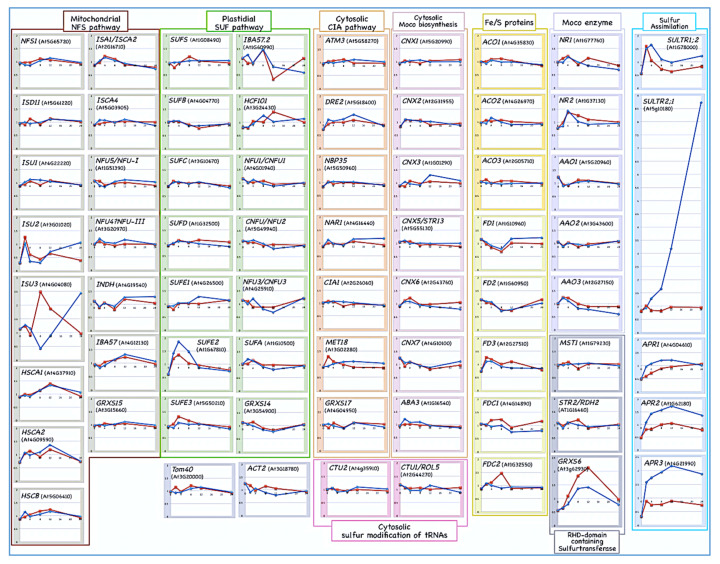
*Arabidopsis* plants of 10-day-old grown on sulfate-containing medium were transferred to the medium with (the same concentration of) or without sulfate, and time course values of the transcripts were monitored for 24 h after sulfate repletion or depletion from the medium. Transcript levels of the genes related to the mitochondrial NFS, plastidial SUF, and cytosolic CIA pathways for Fe/S cluster biosynthesis and to some Fe/S proteins (aconitases and ferredoxins) are indicated. Gene expression levels of the transcripts of the cytosolic sulfur transfer pathways for Moco biosynthesis and sulfur modification of tRNAs also shown. Gene expression levels for some Moco-enzymes (two nitrate reductases *NR1* and *NR2*, and three aldehyde oxidases *AAO1, AAO2,* and *AAO3*), some RHD-domain containing proteins such as mercaptopyruvate sulfurtransferases (*MST1* and *STR2/RDH2*) and a cytosolic glutaredoxin, *GRXS6*, are also indicated. Transcript levels of several *SULTR*s and *ARP*s all of which are involved in the sulfur assimilation process are indicated as the highly responsive ones to the sulfate depletion [80]. Transcript levels of both *TOM40* and *ACT2* are also indicated because they are not related to the sulfur transfer systems mentioned here. All the microarray data used here are described in the previous paper [80]. Transcript levels were analyzed in sulfur-repleted (red line) or in sulfated-depleted (blue line) conditions. For each gene investigated here, gene name, and AGI code numbers are indicated on the top of the column.

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
