# Peer review of "Biosynthesis of Sulfur-Containing Small Biomolecules in Plants"

_ijms, 2020, doi:10.3390/ijms21103470_

Round 1

Reviewer 1 Report

Manuscript ID: ijms -782886

The manuscript entitled “Biosynthesis of sulfur-containing small biomolecules in plants” is a review summarizing the information from recent studies on the biosynthesis pathways of several sulfur-containing small biomolecules and the proteins participating in these processes.

The review starts with a brief description of cysteine biosynthesis, because cysteine is the sulfur donor in the biosynthetic processes of a variety of small S-containing small biomolecules, namely molybdenum cofactor, iron-sulfur clusters and sulfur-containing nucleosides of tRNAs. The authors explain why they have focussed on these molecules and their interconnections as well as the associated intracellular sulfur transfer processes.

The first unit is devoted to iron-sulfur clusters, i.e. the prosthetic groups consisting of labile sulfur and non-heme iron, which are attached onto various apoproteins to form the iron-sulfur protein. The action of cysteine desufurase is explained. Then, two distinct Fe-S cluster biosynthetic patways are described, the mitochondrial versus the plastidial pathway. A cartoon summarizes both pathways.

Apart from the mitochondrial and plastidial proteins that contain Fe-S clusters, there are cytosolic such proteins, too. Therefore, the maturation of the Fe-S cluster and the related proteins in the cytosol are presented in the following unit. Namely, this unit of the text discuss the cytosolic Fe-S cluster assembly pathway, along with two sulfur transfer pathways mediated by the rhodanese domain-containing sulfotransferases that end to the formation of molybdenopterin or sulfur-modified transfer RNAs. A cartoon provides the flow of the various enzymes and their interactions within each pathway.

The next unit elaborates on the sulfur modified Moco, the prosthetic group in which molybdenum is chelated to sulfur atoms bound to the pterin-based frame, as well as on sulfur modification of cytoplasmic tRNAs.

Then, an array of gene expression dynamics related to the biosynthesis of sulfur containing biomolecules in sulfate deprived roots is presented. This is a meta-data analysis of microarray data described in detail in a previous paper of one of the authors. In this analysis the authors have re-analyzed the data and searched for the sulfur-depletion responsive genes involved in the aforementioned sulfur-transfer pathways.

In my opinion, the manuscript is a well written one and reach in information, based on 82 relevant references. In fact, this manuscript nicely summarizes the corresponding network based on works in Arabidopsis. Therefore, it can serve also as a nice platform for similar studies in other plants species, in order to find out whether the presented broad picture is the same or differs in other plant species or families.

L.152-153, I think that the sentence needs improvement.

Author Response

Response to Reviewer 1 Comments,

Point : L.152-153, I think that the sentence needs improvement.

Response: According to the Reviewer’s suggestion, I changed the sentence pointed out at L.161-L.162.

Reviewer 2 Report

  In this manuscript, Nakai and Maruyama-Nakashita review the knowledge of sulfur metabolism in plants, with a particular emphasis on sulfur containing molecules such as FeS, MoCo and thio-modified tRNAs. The review is generally well written and should be appealing to the broad readership of Int. J. Mol. Sci. Below are my suggestions for improvement of the manuscript.
1) Is sulfur part of a cycle like Nitrogen (going from N2 to nitrogenous compounds and back to N2)? If so, it might informative to indicate this in the manuscript.
2) What is the content in sulfur of a plant cell and where is most of the sulfur allocated in the cell. If such an analysis has been done, it might be worthwhile to report the findings here.
3) Can the authors specify where the different steps of cysteine synthesis take place in the plant? Is cysteine synthesized in the plastid stroma? If so, there must exist in the chloroplast membrane transporters for sulfide and cysteine exporters. I am assuming cysteine importers at the mitochondrial inner membrane must also operate. Have they been described?
4) As the authors devote significant space to FeS cluster biogenesis, they might want to include a table listing all the known FeS containing proteins in organelles and elsewhere in the cell.
5) It is unclear to me if FeS made in the mitochondria can be exported to be delivered to target proteins? Can the authors clarify what is known about this in the field?
6) Are sulfur modifications of tRNAs in mitochondria and plastids also taking place?
7) There is no mention of GSH synthesis and considering its essential role in plant metabolism, the authors might want to add a couple of sentences and refer the reader to additional reviews if they don’t wish to expand on this particular topic.
8) The conclusion seems to be an expansion of the information described in the other sections. I would suggest the authors provide a succinct conclusion emphasizing the pending questions in the field.
9) Are there mechanisms to recycle the sulfur in macromolecules into sulfate?

Author Response

Response to Reviewer 2 Comments,

Point 1: Is sulfur part of a cycle like Nitrogen (going from N2 to nitrogenous compounds and back to N2)? If so, it might informative to indicate this in the manuscript.

Response 1: In this review, we have focused on the intracellular sulfur transfer events in plants. Therefore the sulfur recycling system in the environment may be peripheral to the focus of this review. In general, an organic sulfur compound dimethylsulphoniopropionate (DMSP), which is a secondary metabolite produced by phytoplankton, is known to evaporate into the atmosphere as sulfuric acid and organic aerosols and return to land and sea as rainfall [ref. 2].

Point 2: What is the content in sulfur of a plant cell and where is most of the sulfur allocated in the cell. If such an analysis has been done, it might be worthwhile to report the findings here.

Response 2: According to the Reviewer’s suggestion, we have added some information about this in the Introduction session (L.30-L.32).

Point 3: Can the authors specify where the different steps of cysteine synthesis take place in the plant? Is cysteine synthesized in the plastid stroma? If so, there must exist in the chloroplast membrane transporters for sulfide and cysteine exporters. I am assuming cysteine importers at the mitochondrial inner membrane must also operate. Have they been described?

Response 3: Conversion into sulfide occurs in plastids, and, the subsequent reactions including cysteine biosynthesis occur not only in plastids but also in the cytosol and mitochondria. To clarify this point, we have added this information at L.36-L.39.

The sulfide-specific exporter has not been identified so far. Amino acid transporters are known to have a broad specificity to various amino acids so that the cysteine-specific transporter has not been known at present.

Point 4: As the authors devote significant space to FeS cluster biogenesis, they might want to include a table listing all the known FeS containing proteins in organelles and elsewhere in the cell.

Response 4: Actually a large number of the Fe/S proteins are found in both organelles in all phyla of organisms including plants. References in which describe those plant Fe/S proteins in details have been added at L.90.

Point 5: It is unclear to me if FeS made in the mitochondria can be exported to be delivered to target proteins? Can the authors clarify what is known about this in the field?

Response 5: As described in the first paragraph of the section 3 (L.128-L.138), no evidence has been demonstrated so far that the Fe/S cluster itself is directly exported from mitochondria. However, as already described in the text, as the mitochondrial Fe/S cluster biosynthesis is absolutely required for cytosolic Fe/S cluster maturation, an yet-unknown sulfur compound should be exported from mitochondria to be utilized for cytosolic Fe/S cluster formation.

Point 6: Are sulfur modifications of tRNAs in mitochondria and plastids also taking place?

Response 6: The actual existence of organelle tRNA sulfur modifications in plants has not yet been clarified, and we are currently trying to analyze this point using Arabidopsis thaliana.

Point 7: There is no mention of GSH synthesis and considering its essential role in plant metabolism, the authors might want to add a couple of sentences and refer the reader to additional reviews if they don’t wish to expand on this particular topic.

Response 7: According to the Reviewer’s suggestion, we have added some information of GSH biosynthesis and its cellular function with additional references at L.51-L.53.

Point 8: The conclusion seems to be an expansion of the information described in the other sections. I would suggest the authors provide a succinct conclusion emphasizing the pending questions in the field.

Response 8: According to the Reviewer’s suggestion, we have changed the title of this section from “6. Conclusion” to “ 6. Perspectives’, and we have condensed the text slightly to clarify the remaining open questions in the field.

Point 9: Are there mechanisms to recycle the sulfur in macromolecules into sulfate?

Response 9: We believe that such recycling mechanisms should exist in plants but recycling of sulfur from sulfur-containing proteins to sulfate has not yet been elucidated in details.